🔓 | **Open Peer Review** | Genetics and Molecular Biology | Research Article

# Phosphoglucomutase A-mediated metabolic adaptation is essential for antibiotic and disease persistence in *Mycobacterium tuberculosis*

Taruna Sharma,[1,2] Shaifali Tyagi,[1,2] Rahul Pal,[1] Jayendrajyoti Kundu,[3] Sonu Kumar Gupta,[4] Vishawjeet Barik,[1,2] Vaibhav Kumar Nain,[1,2] Manitosh Pandey,[1] Prabhanjan Dwivedi,[5] Bhishma Narayan Panda,[5] Yashwant Kumar,[3] Ranjan Kumar Nanda,[6] Samrat Chatterjee,[3] Amit Kumar Pandey[1,5]

**ABSTRACT**  The long-term survival of *Mycobacterium tuberculosis* (Mtb) requires efficient use of host resources and uninterrupted access to host-derived nutrients. This is done by utilization of a highly flexible and integrated network of metabolic pathways. Phospho-glucomutase A (*pgmA*) is essential for glycogen biosynthesis, which acts as a nutrient reservoir and is known to modulate carbon flux in various pathogens. We, for the first time, investigated the role of *pgmA* in Mtb by creating a strain lacking this gene. The absence of *pgmA* hinders the survival of pathogens under nutrient-limiting and reactivation conditions. Our study shows that the lack of cell membrane-associated glycolipids in Δ*pgmA* compromises cell wall integrity and increases susceptibility to stress. Interestingly, Δ*pgmA* exhibits an enhanced growth phenotype on cholesterol compared to the wild type due to low cyclic adenosine monophosphate (cAMP) levels. Differential gene expression and $^{13}C_3$ carbon dilution analyses indicate that stored carbon as glycogen is crucial for Mtb survival under nutrient-limiting conditions. We demonstrate that *pgmA* is vital for Mtb growth within the host. This study highlights the critical role of *pgmA* in metabolic adaptation during nutrient starvation and reactivation and its implication on antibiotic and disease persistence. These insights are crucial for developing novel, shorter, and more effective anti-tuberculosis strategies.

**IMPORTANCE**  This study for the first time investigated the role of metabolic enzyme phosphoglucomutase A (*pgmA*) in *Mycobacterium tuberculosis* (Mtb), revealing its crucial functions as a toggle switch between biosynthesis and growth. This work highlights the importance of *pgmA* in maintaining the metabolic flexibility of Mtb during the nutritional switch. The presence of *pgmA* is critical for the production of membrane-associated glycolipid, which helps maintain the cell wall integrity under various growth and stress conditions. This adaptability is pivotal for generating starvation-induced antibiotic tolerance in Mtb. In addition to the clinical context, these findings provide a mechanistic foundation for understanding adaptive strategies by Mtb to harsh environments and the development of drug-tolerant bacilli.

**KEYWORDS**  *Mycobacterium tuberculosis*, phosphoglucomutase A, *pgmA*, glycogen metabolism, virulence, antibiotic tolerance, persistence, trehalose

Metabolic plasticity is essential for *Mycobacterium tuberculosis* (Mtb) to adapt to changing environments and maintain structural stability despite fluctuations in nutrient availability. This adaptability is particularly crucial for the survival of the pathogen within the nutrient-deprived intracellular niches of the host (1). To counter this, Mtb has evolved strategies that regulate the carbon flux for an efficient utilization of

**Peer Reviewer** Samuel Alvarez-Arguedas, UT Southwestern Medical Center, Dallas, Texas, USA

Address correspondence to Amit Kumar Pandey, amitpandey@thsti.res.in.

Shaifali Tyagi and Rahul Pal contributed equally to this article.

The authors declare no conflict of interest.

See the funding table on p. 20.

available resources. The continuous metabolic shift between growth and biosynthesis is tightly regulated by a network of metabolic and signaling pathways that facilitate this process. Despite playing a major role in the disease pathogenesis, the precise mechanism that maintains this balance remains largely elusive. Phosphoglucomutase (*pgmA*), an enzyme that reversibly converts glucose-6-phosphate (G6P) to glucose-1-phosphate (G1P), is one such enzyme that regulates the flux of the carbon by modulating this transition (2, 3). As and when the need arises, this reversible transformation enables the pathogen to regulate the flow of carbon between various catabolic and biosynthesis pathways. The catabolic pathway originating from G6P is known to support growth by providing energy. Alternatively, during biosynthesis, the downstream metabolites generated using G1P essentially contribute toward the synthesis and maintenance of the lipid-rich cell wall. However, *pgmA*-dependent regulation of the central carbon metabolism (CCM) and its role in tuberculosis (TB) disease biology remains relatively unexplored.

In certain bacterial species, the inhibition of the *pgmA* gene has been associated with decreased energy production and diminished bacterial survival, particularly during shifts in carbon flux within the host environment (4, 5), cell wall biosynthesis, and antibiotic resistance (6–11). Non-pathogenic Δ*pgmA* strains from *Brucella melitensis* and *Streptococcus iniae* have shown *pgmA* to be a promising candidate for the development of attenuated vaccine strains (12, 13). Metabolic products from *pgmA*-dependent G1P arm include crucial intermediates such as trehalose, UDP-glucose, maltose-1-phosphate, and finally glycogen, whose role in membrane biogenesis and integrity is well known (14–18). Glycogen is a universally conserved storage molecule and is known to act as a carbon reservoir during nutrient scarcity (19). In *Mycobacterium smegmatis* (Msm), it was reported that in addition to the role of glycogen as a conventional storage macromolecule, a continuous synthesis and degradation of glycogen throughout the exponential growth phase happen, suggesting its involvement as a carbon capacitor for glycolysis during active growth (20). Although we have some information on the role of glycogen metabolism in the growth and survival of Msm, its role in mycobacterial drug susceptibility, virulence, and pathogenesis is lacking.

In this study, we aimed to elucidate the role of *pgmA* in maintaining carbon flux balance during nutrient scarcity in Mtb. To accomplish this, we generated a genetically modified Mtb strain lacking the *pgmA* gene (Δ*pgmA*). We found that relative to the wild type (WT), Δ*pgmA* demonstrated a decrease in their ability to grow under both nutrient-limiting and reactivation conditions. The absence of *pgmA* in Mtb resulted in reduced generation of glycogen, trehalose, and other membrane-associated glycolipids, compromising both the cell wall integrity and the ability to survive under stress. Intriguingly, we found that *pgmA*-mediated growth modulation on cholesterol is one of the critical drivers of antibiotic and disease persistence in tuberculosis. Understanding the intricacies of central carbon metabolism in mycobacteria holds potential for innovative therapeutic strategies aimed at disrupting their energy equilibrium and virulence. Furthermore, targeting mycobacterial *pgmA* could provide a promising approach for developing tailored antimicrobial agents to combat mycobacterial infections.

## RESULTS

### *pgmA* is essential for the survival of Mtb under nutrient-limiting conditions

*pgmA* is an isomerase involved in the inter-conversion of G1P to G6P (Fig. 1A), a critical step involved in the synthesis of metabolites fundamental to CCM. To further characterize this gene, we generated the *pgmA* gene deletion mutant strain in H37Rv by homologous recombination technique (Fig. S1A). Furthermore, the gene deletion in Δ*pgmA* at the gene and transcript levels was confirmed by PCR and RT-PCR methods, respectively (Fig. S1B and C). Additionally, as a control, we also generated the complemented strain by expressing the *pgmA* gene in the mutant strain. As glycogen is the end product of the pathway, we first measured its levels under glucose-free conditions,

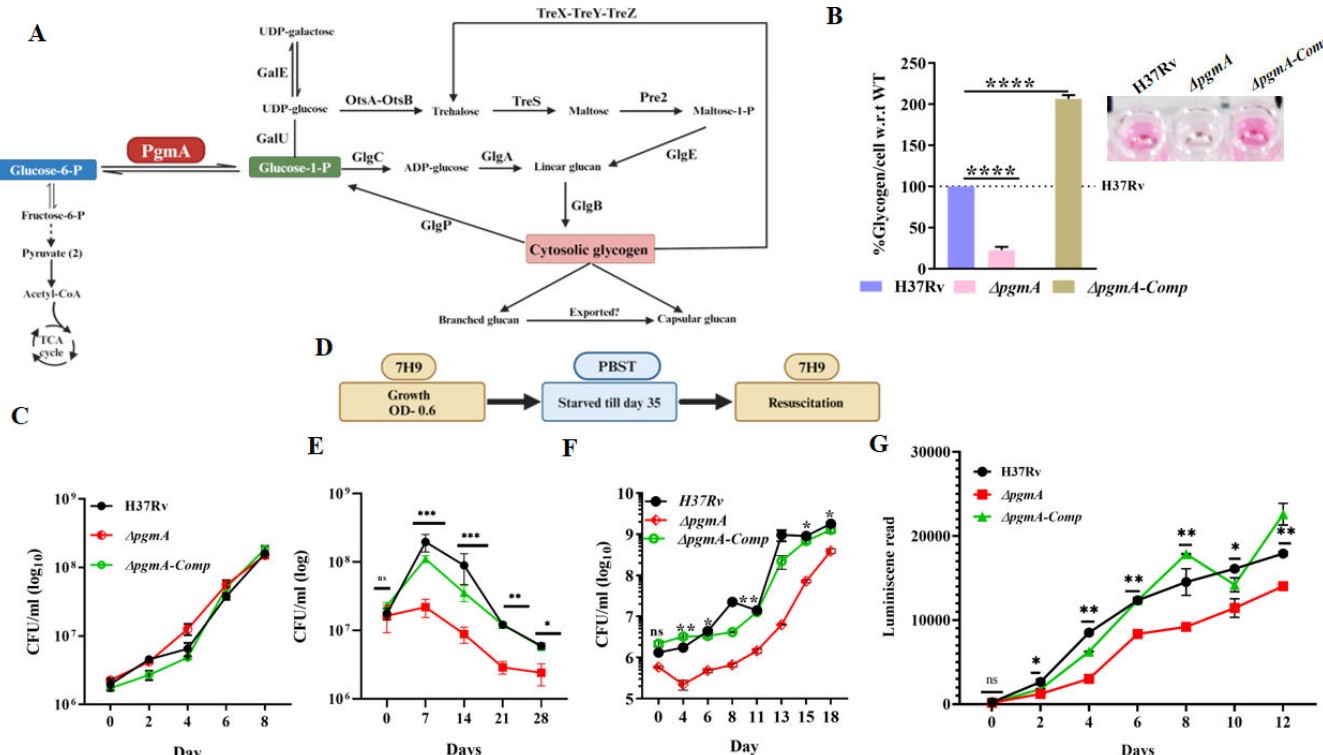

**FIG 1** *pgmA* is essential for the survival of Mtb under nutrient-limiting conditions. (A) Schematic representation of *pgmA* catalyzing the inter-conversion of G6P and G1P depicting the intermediates crucial for biosynthesis and growth. (B) Altered glycogen levels in the absence of *pgmA* at day 7 starvation in PBST. The inset shows the colorimetric estimation of glycogen levels, highlighting visual differences between the strains. (C) Growth kinetics of Δ*pgmA* under standard growth conditions. (D) Illustration outlining the experimental approach for carbon starvation and subsequent resuscitation. (E) Δ*pgmA* fails to survive under carbon starvation and is essential during (F) resuscitation. (G) Resuscitation kinetics were assessed using luminescence-based reporter bacteria, enabling real-time tracking of bacterial recovery dynamics following 4 days of starvation in H37Rv, Δ*pgmA*, and Δ*pgmA-Comp* strains. The luminescence readings for day 0 were normalized by subtracting the background luminescence for all three strains to ensure accurate comparison. Statistical significance, (B) to (C) and (E) to (G), was determined using unpaired, non-parametric, two-tailed *t*-test, *$P \leq 0.05$ and **$P \leq 0.005$. Data represent mean ± standard deviation for technical triplicates.

since the assay kit detects free glucose by enzymatically breaking down the glycogen complex. As expected, we observed a significant decrease in the level of glycogen in the Δ*pgmA* relative to the wild type, which was found to be unaffected in the complemented strain (Fig. 1B, also see Fig. S2A). Despite altered glycogen levels, we did not observe any difference in the ability of Δ*pgmA* to grow under nutrient-sufficient conditions (Fig. 1C). Since glycogen is universally known to act like a nutrient reservoir, we were interested to know its role in providing survival advantage to Mtb under both nutrient-limiting and resuscitation conditions (Fig. 1D). Interestingly, relative to the wild-type and the complemented strains, we observed a significant decline in the ability of Δ*pgmA* to survive under starvation conditions observed in terms of both colony forming unit (Fig. 1E) and total biomass (Fig. S2B). Additionally, the reactivation of the starved Δ*pgmA* in enriched media also demonstrated an initial delay in the resuscitation phenotype in comparison to the wild-type and the complemented strains (Fig. 1F). Furthermore, to minimize the effect of starvation on the overall reactivation process, we repeated the experiment and recorded the reactivation profile starting 4 days post-starvation. Use of the bioluminescent version of wild-type, mutant, and the complemented strains allowed the real-time tracking of bacterial recovery dynamics during reactivation. Luminescence readings, which reflect bacterial viability, on day 0 were normalized by subtracting the background luminescence for all three strains. Our data suggest that Δ*pgmA* exhibits a pronounced resuscitation defect compared to the WT and complemented strains (Fig. 1G). Lower ATP levels were observed under starvation conditions, indicating

compromised fitness and low metabolic activity of ΔpgmA in such environment (Fig. S2C). The above observation suggests that *pgmA* is crucial for the glycogen biosynthesis pathway and facilitates the survival fitness of Mtb under both nutrient-limiting and resuscitation conditions.

## *pgmA* is crucial for maintaining the cellular architecture of Mtb

*pgmA*-dependent synthesis of G1P acts as a precursor of the biosynthesis of critical metabolites essential for the synthesis and transportation of cell wall-associated sugars and lipids (21–24). We hypothesize that the absence of *pgmA* will significantly compromise the membrane integrity of the Mtb. The same was first tested by performing the ethidium bromide (EtBr) permeability assay. High EtBr uptake was observed by ΔpgmA relative to the wild-type and the complemented strains (Fig. 2A), suggesting that the absence of *pgmA* enhances the cell permeability by altering the integrity of the cell membrane. Furthermore, total lipids extracted from cultures were separated using thin-layer chromatography (TLC) technique. Total lipid was separated on a silica plate (stationary phase) using different combinations of the organic solvents (mobile phase) (Fig. S3A). Surprisingly, TLC analysis of membrane lipids revealed a significant reduction in band intensity of cell wall-associated lipids, including triacylglycerol (TAG), phthiocerol dimycocerosates (PDIM) (Fig. 2B; Fig. S3B), and mycolic acids (Fig. 2C), in ΔpgmA relative to the wild type.

To understand the morphological implications of these lipid abnormalities on ΔpgmA, we visualized these strains using transmission electron microscopy (TEM) (Fig. 2D). We found that the absence of the *pgmA* gene resulted in an increase in the diameter (Fig. 2E)

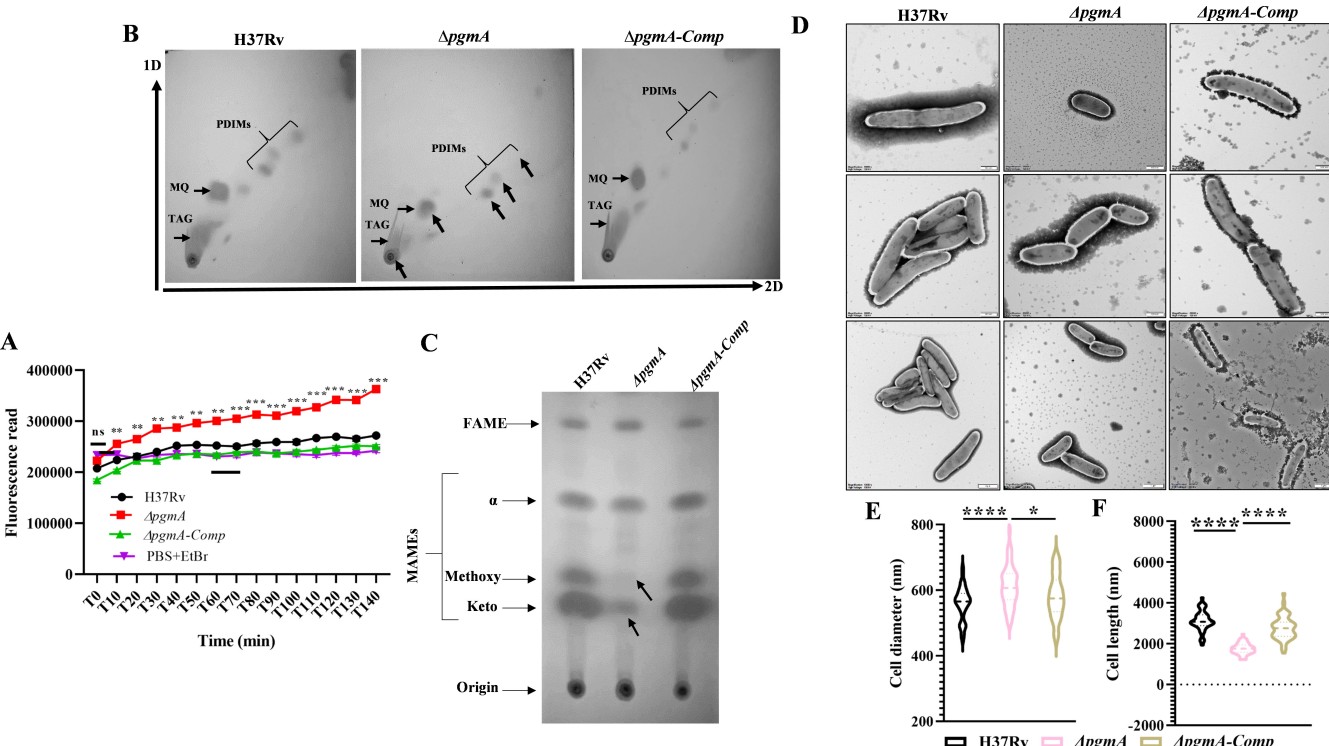

**FIG 2** *pgmA* is crucial for maintaining the cellular architecture of Mtb. (A) Increased membrane permeability was observed in the ΔpgmA strain. In (B) to (C), total lipids were separated by growing bacterial cells until A$_{600}$ reached 1–1.5. TLC reveals deficiencies in (B) PDIM-A, TAGs, and (C) mycolic acid in ΔpgmA. 2D lipid separation was achieved using hexane:ethyl acetate (98:2) for the first dimension and hexane:acetone (98:2) for the second dimension. In (D), transmission electron microscopy images post 7 days of starvation depict (E) an increase in overall cell diameter and (F) a decrease in cell length (*n* = 50 cells each strain) of ΔpgmA. The cells were visualized at 20,000× to 12,000× magnification (top to bottom) on a 500 nm to 1 µm scale (top to bottom) at 120 kV. In (E) and (F), the cell size measurement was done using RADIUS 2.0 EMSIS software. Statistical significance, (A), (E), and (F), was determined using unpaired, non-parametric, two-tailed *t*-test, *$P \leq 0.05$ and ****$P \leq 0.00005$. FAMEs, fatty acid methyl esters; MAMEs, mycolic acid methyl esters; MQ, menaquinone.

with a simultaneous shortening in the length of individual bacteria (Fig. 2F). This resulted in imparting a more spherical shape to the mutant relative to the elongated morphology observed in the wild-type and complemented strains. Interestingly, in comparison to the wild-type and complemented strains, defects in overall membrane lipid biosynthesis significantly impacted the ability of ΔpgmA to form biofilm (Fig. S3C). The lipid isolated from the biofilms suggests the said effects were found to be associated with a deficiency in TAGs, PDIMs, and trehalose mycolates (Fig. S3D). This observation was linked by evaluating the minimum inhibitory concentration (MIC) of vancomycin, wherein, as expected, the ΔpgmA strain exhibited a significantly reduced Half Maximal Inhibitory Concentration (IC50) in response to the drug treatment (Fig. S3E). The observational data presented above underscore the pivotal role of pgmA in preserving the integrity of the membrane and the overall cell structure of Mtb.

## Deletion of pgmA renders Mtb susceptible to various stresses

To assess the role of pgmA in the growth of Mtb inside the cell, we analyzed the fitness of ΔpgmA by exposing different Mtb strains to host-induced stressors (Fig. 3A). After 48 h, ΔpgmA exhibited approximately ~50% susceptibility to nitrosative stress. Sensitivity levels were approximately ~90% for oxidative stress, ~60% for pH stress, and ~80% for hypoxic stress. Based on the above finding, we hypothesized that the pgmA gene of Mtb must be playing a crucial role in helping Mtb survive inside the macrophages. To test this, we further estimated the relative fitness of the wild-type, ΔpgmA, and complemented strains to replicate inside macrophages. Briefly, as depicted in Fig. S4A, both resting and activated Human Acute Monocytic Leukemia Cell Line (THP-1) were infected by giving a 1:1 multiplicity of infection (MOI), and on the 6th day post-infection, THP-1 cells were lysed and the intracellular bacteria were assessed by colony-forming unit (CFU) plating. We found that relative to the wild-type and the complemented strains, ΔpgmA demonstrated reduced fitness in its ability to survive inside both resting and Interferon-gamma (IFN-γ) activated macrophages (Fig. 3B). Note the percent uptake was observed to be equal, ensuring accurate infection efficiency (Fig. S4B). Consistent with the CFU assay results, we found that activated THP-1 macrophages infected with the ΔpgmA mutant strain showed increased phagosome-lysosome fusion compared to wild-type and complemented strain-infected macrophages (Fig. 3C and D). The observed attenuation in the THP-1 macrophage infection assay correlates with the increased co-localization of the ΔpgmA strain to the acidic compartments within the host cell. This enhanced localization to the acidic compartment suggests that the mutant strain is more susceptible to the intracellular stress imposed by macrophages compared to the WT and complemented strains.

## pgmA-mediated transcriptional reprogramming helps Mtb survive under nutritionally limiting conditions

To further understand the role of pgmA in supporting the growth of Mtb during starvation, we conducted differential gene expression analysis studies to identify genes and pathways essential for pgmA-mediated survival during starvation. The transcriptional analysis began with an initial comparison of respective transcriptomic profiles of both H37Rv and ΔpgmA in starved versus normal conditions. To specifically identify genes differentially expressed by pgmA during starvation, we excluded the starvation-specific signal common to both H37Rv and ΔpgmA from our final analysis (Fig. 4A). In the final analysis, 26 pgmA-specific genes were found to be significantly upregulated (red dots), whereas 198 genes were downregulated (blue dots) under starvation conditions. (Fig. 4B).

Notably, Rv3290, a well-established global regulator of mycobacterial persistence, emerged as the most up-regulated gene in the ΔpgmA strain under starvation conditions (28, 29). Additionally, several genes encoding membrane-associated proteins were identified in the up-regulated gene set of ΔpgmA strain differentially expressed genes (DEGs), including Proline–Glutamate Polymorphic GC-Rich Sequence (PE-PGRS; Rv2340c)

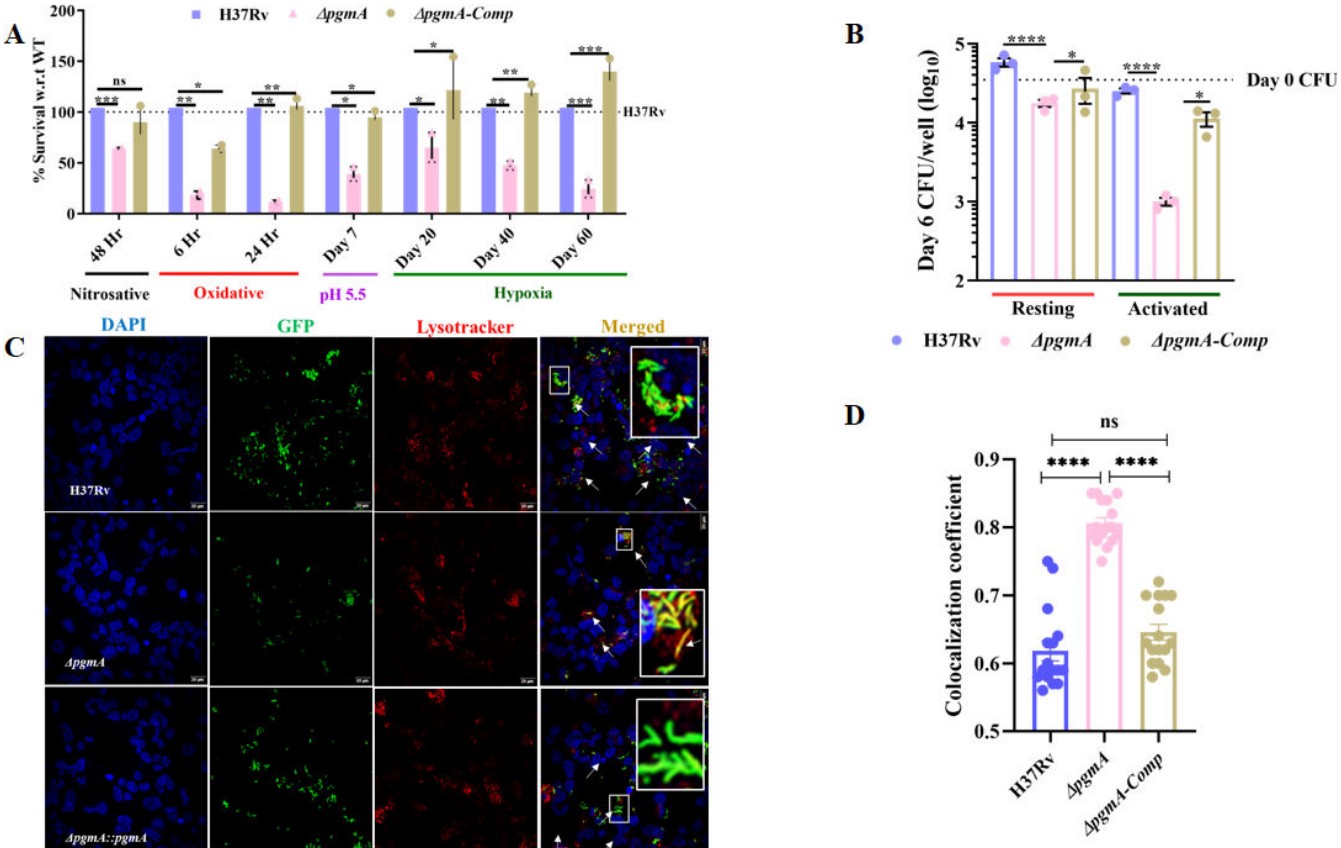

**FIG 3** Deletion of *pgmA* renders Mtb susceptible to various stresses. (A) H37Rv, Δ*pgmA*, and complemented strains were subjected to different stress conditions: nitrosative stress with 200 µM of DETA-NO for 48 h and oxidative stress with 5 mM of $H_2O_2$ treatment for 6 h. The percent survival of Δ*pgmA* relative to the wild-type strain was calculated by plating cultures at day 0 and respective time points. (B) Resting and IFN-γ-activated (10 ng/mL) THP-1 cells infected with 1:1 (MOI) of infection with H37Rv and Δ*pgmA*. In (C) and (D), the deletion of *pgmA* was observed to enhance the co-localization of Mtb in the acidic compartments of THP-1 macrophages. In (C), confocal imaging of THP-1 cells stained with 4′,6-diamidino-2-phenylindole (DAPI; blue panel), bacteria having over-expressed GFP levels (green panel), and LysoTracker staining the acidic compartments (red panel). Activated THP-1 macrophages were infected with Green Fluorescent Protein (GFP)-labeled strains, and images were captured using an Olympus FV3000 confocal microscope at 60× magnification (scale bars, 20 µm). Yellow region shows GFP-tagged bacterial population in the acidic compartment (red stained) of macrophage. In (D), the co-localization coefficient between red and green fluorescent signals for all three strains was determined using Olympus cellSens software. The data presented for co-localization coefficient were calculated using 16 z-stacked images obtained from two independent experiments. Statistical significance, (A) to (B) and (D), was determined using unpaired, non-parametric, two-tailed *t*-test, *$P < 0.05$, **$P < 0.005$, ***$P < 0.0005$, and ****$P ≤ 0.00005$, ns, not significant. Data represent mean ± SEM for technical triplicates.

(30), a stress-response membrane protein (Rv1004c) (31), and permeases (*Rv1999c*) (32–34) that are potentially involved in L-arginine uptake. Furthermore, some of the up-regulated metabolic enzymes, such as Rv0356c (a putative thioesterase) and Rv0223c (an aldehyde dehydrogenase), are known to play a crucial role in responding to oxidative stress conditions, particularly those that induce membrane lipid peroxidation (35), and further underscore the intricate link between *pgmA* and bacterial adaptation to starvation-induced stress. Conversely, the list of genes that was down-regulated included genes with metabolic and respiratory functions. Notable genes within this cluster include *canB* (*Rv3588c*), *pcaA* (*Rv0470c*), *hemE* (*Rv2678c*), *cyp123* (*Rv0766c*), *mbtD* (*Rv2381c*), *murG* (*Rv2153c*), *ppsD* (*Rv2934*), *glmS* (*Rv3436c*), *pfkA* (*Rv3010c*), *lipY* (*Rv3097c*), *lepA* (*Rv2404c*), *fadD2* (*Rv2948c*), *ppsB* (*Rv2932*), *gpdA2* (*Rv2982c*), *fabG5* (*Rv2766c*), *rsfB* (*Rv3687c*), *fadD22* (*Rv2948c*), and *fadD35* (*Rv2505c*). The list further included genes that encoded membrane proteins *Rv0666*, *Rv0488*, *Rv0219*, *Rv1382*, *Rv2254c*, *Rv0680c*, *Rv2307c*, *Rv0463*, *Rv0048c*, *Rv2293c*, *Rv0289* (*espG3*), *Rv2723*, *Rv2686c* (ABC transporter), *Rv2325c*, *Rv2403c* (*lppR*), *Rv1986*, *Rv2643* (*arsC*), *Rv3454*, *Rv0473*, *Rv3821*, *Rv1541c*, *Rv1217c* (ABC transporter),

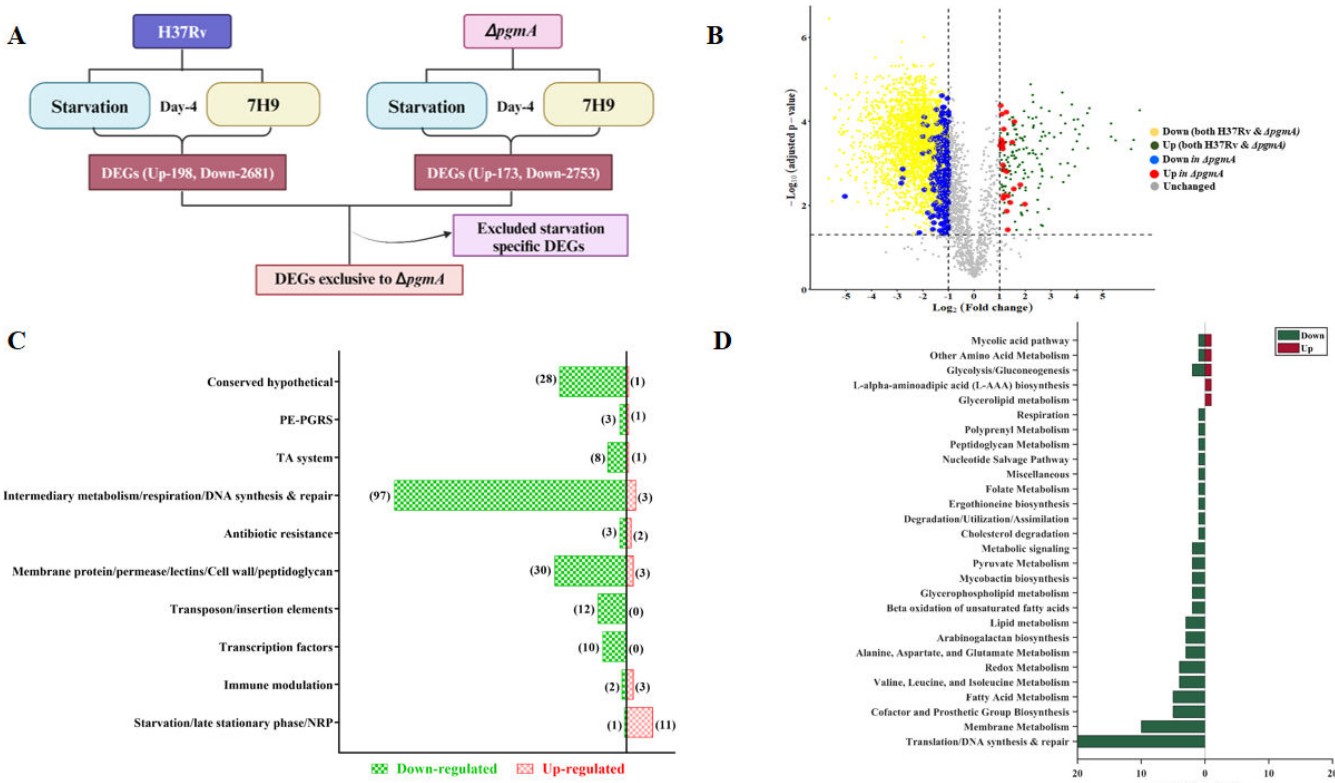

FIG 4 *pgmA*-mediated transcriptional reprogramming helps Mtb survive under nutrient-limiting conditions. (A) Schematic representation of strategy involved in RNAseq sample processing and analysis. (B) The volcano plot illustrates the results of a differential expression analysis comparing gene expression in starving versus normal conditions for both H37Rv and *ΔpgmA*. Differentially expressed genes (DEGs) were identified using a log$_2$ fold change cut-off of 1 and a false discovery rate *P*-value <0.05 (25). The genes that are exclusively up- and down-regulated in *ΔpgmA* during starving conditions are highlighted in red and blue, respectively. Commonly up- and down-regulated DEGs in both H37Rv and *ΔpgmA* are depicted in green and yellow, respectively. In (C), functional characterization of the uniquely regulated genes extracted using gene ontology in *ΔpgmA* is presented (Table S3). In (D), the bars represent the count of significantly altered metabolic genes involved in enriched metabolic pathways specifically for *ΔpgmA* during starving conditions. Pathway information was extracted from the genome-scale metabolic model of Mtb H37Rv, known as iEK1011 (26), and the BioCyc genome database (27). Herein, the red and green colors of the bars represent the count of up- and down-regulated genes, respectively (Table S4). Full forms;Proline–Glutamate Polymorphic GC-Rich Sequence - PE-PGRS

*Rv1146* (*mmpl13b*), *Rv1038c* (*esxJ*), *Rv1793* (*esxN*), *Rv1914c, Rv2620c, Rv0588* (*yrbE2B*), *Rv0008c*, and *Rv0584*, which indicates severe membrane remodeling associated with *ΔpgmA* strain under nutrient scarcity. Functional characterization of these genes using gene ontology revealed that a majority of the gene subset up-regulated belong to pathways critical for maintaining the late stationary phase, non-replicating persistence, and survival of Mtb during starvation (Fig. 4C; Table S3). Since the *pgmA* gene encodes for an enzyme that critically controls the flux of carbon flow between energy and biosynthesis (Fig. 1A), the absence of the *pgmA* gene would certainly destabilize the CCM of Mtb essential for regulating optimal energy production and biosynthesis during starvation. To study this imbalance, we extracted the metabolic pathways using the metabolic gene and its associated metabolic reactions from the metabolic model of Mtb iEK1011_2.0 and the BioCyc genome database as described previously (27, 36). The bars in the graph depict the count of significantly altered metabolic genes (*x*-axis) involved in enriched metabolic pathways (*y*-axis) specifically for *ΔpgmA* during starvation (detailed in Table S4). The coloration of the bars, red and green, represents the number of up-regulated and down-regulated metabolic genes, respectively. A substantial down-regulation was observed in the majority of metabolic pathways, including 97 metabolic genes associated with membrane metabolism, fatty acid metabolism, nucleotide metabolism, and other critical pathways (Fig. 4D). These data suggest that *pgmA*

modulates the metabolic transcriptome in a way that helps Mtb to adapt and survive better under nutritional deprivation.

## *pgmA* is crucial for the metabolic adaptability of Mtb under nutrient-limiting conditions

To investigate the regulation of carbon flux in response to nutrient scarcity, we conducted a $^{13}$C carbon dilution tracing experiment using uniformly labeled U-$^{13}$C$_3$ glycerol (Gly). The objective was to investigate the rate of change of *pgmA*-dependent exhaustion in the $^{13}$C$_3$ pool in the major metabolic flux when Mtb faces nutrient-limiting conditions under impaired glycogen levels. To achieve this, both H37Rv and the Δ*pgmA* strains were first grown until mid-log phase OD: 0.6 in minimal media supplemented with labeled U-$^{13}$C$_3$ glycerol. Subsequently, to induce a starvation response, the spent media were replaced with PBST, and the strains were further cultured for 7 days to precisely track carbon utilization and distribution within CCM during nutrient deprivation and to get better insights into how the strains will adapt at metabolite levels under nutrient-depleted conditions over time.

Samples were processed, and liquid chromatography–mass spectrometry (LC-MS) data acquisition was done at different time points (Fig. S5A). It is well-established that $^{13}$C$_3$ glycerol is directed from the glycolytic flux through glycerate-3-phosphate (37). Initially, at time point zero, just before the onset of starvation, we identified a significant shift in $^{13}$C$_3$ flux, particularly evident in trehalose and maltose-1-phosphate levels, indicating impaired G1P to G6P conversion and corresponding downstream metabolic disturbance (Fig. 5A). Following starvation, as anticipated, there was a notable impairment in glycolytic flux, as evidenced by the decreased levels of G6P and Fructose-6-Phosphate (F6P), along with a reduction in the levels of intermediates in the tricarboxylic acid (TCA) cycle in the *pgmA*-deficient strain. Conversely, trends in the biosynthetic arm, which includes trehalose and maltose-1-phosphate, remained relatively steady. A significant decrease in $^{13}$C$_3$ labeling was also observed in the amino acid pool such as tryptophan, glutamate, alanine, proline, phenylalanine, isoleucine, and tyrosine (Fig. 5B). Interestingly, the decrease in cAMP levels observed in in-vitro colorimetric-based estimation (Fig. S5B) was further confirmed by a relative decrease in the $^{13}$C$_3$ labeling of the cAMP isolated from in Δ*pgmA* (Fig. 5B). Taken together, the above data suggest that the carbon stored in the form of glycogen under nutrient-replete conditions helps Mtb to maintain the CCM when exposed to a stringent nutrient-limiting niche inside the host (Fig. 5C).

## *pgmA* is essential for cholesterol-specific modulation of growth and antibiotic persistence in Mtb

The role of cholesterol in the persistence and survival of mycobacteria is well-documented (38–40). Earlier Mtb transposon mutant library studies have predicted that the absence of *pgmA* imparts a growth advantage phenotype to Mtb in media having cholesterol as a sole carbon source (Fig. S6A) (41). To further confirm this finding, we performed the growth kinetic studies and assessed the growth of Δ*pgmA*, wild-type, and complemented strains in media harboring either glycerol or cholesterol as a sole carbon source. As previously reported, we also found that in comparison to the wild-type and the complemented strains, Δ*pgmA* demonstrated an enhanced growth rate (~3–5-fold) which was specific to media containing cholesterol as a sole carbon source (Fig. 6A). ATP, being the currency of energy, has a direct bearing on the growth rate of living beings. While there was no difference in the ATP levels between strains grown on glycerol, we observed that Δ*pgmA* grown on cholesterol was presented with more than threefold and sixfold high levels of ATP relative to the parental and complemented strains, respectively (Fig. 6B). Since it is well documented that the cAMP levels inside the pathogen are inversely correlated with the rate of growth of Mtb in cholesterol, we quantified the total cAMP levels in all three strains (Fig. 6C). While we did not find any difference in the cAMP levels in all three strains in glycerol, surprisingly, the cAMP levels in cholesterol media

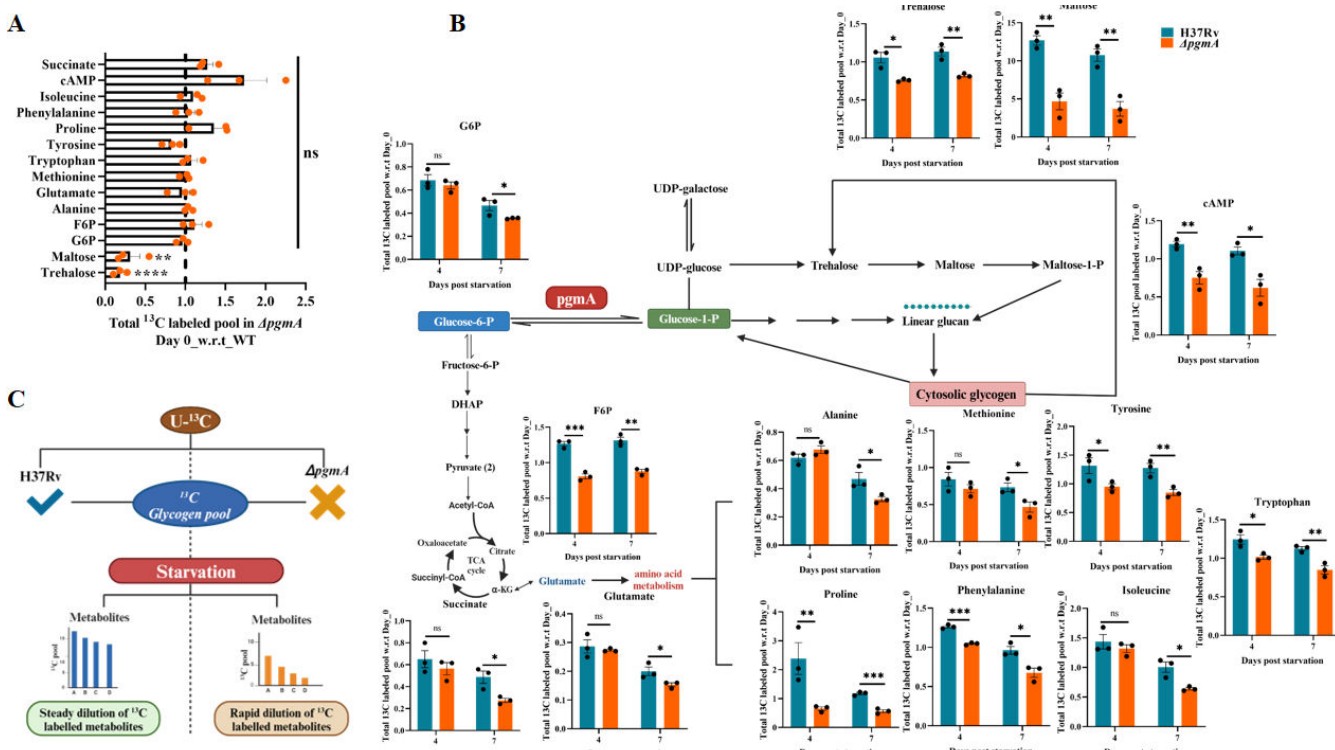

**FIG 5** *pgmA*-mediated regulation is crucial for the survival of Mtb under nutrient stress. (A) The initial abundance of U-$^{13}$C$_3$-labeled metabolites in the pool before the onset of starvation, relative to H37RV, at time point zero. (B) Each bar for H37Rv (blue) and Δ*pgmA* (orange) represents the ratio of the total pool $^{13}$C$_3$ labeled for individual metabolite at day 4 and 7 post-starvation. This includes metabolic lineation from both G6P and G1P. (C) *pgmA* tightly regulates the carbon flux between G6P and G1P during nutrient starvation. All labeled abundances were normalized to the CFU counts of viable bacteria at days 0, 4, and 7. Statistical significance between the ratio of $^{13}$C-labeled pools at days 4 and 7 with respect to (w.r.t.) day 0 for each metabolite was analyzed using unpaired *t*-test, *$P < 0.05$, **$P < 0.005$, ***$P < 0.0005$, and ****$P \leq 0.00005$; ns, not significant. Gly-3-P, glycerate-3-phosphate; α-KG, α-ketoglutarate; C, carbon. Data represent mean ± SEM.

were found to be different in all three strains. While we observed a ~3- and ~16-fold increase in the levels of cAMP in H37Rv and the complemented strain (as compared to glycerol), surprisingly, a ~ 5.7-fold decrease was observed as compared to the parent, and a ~14-fold decrease was observed as compared to the glycerol media, respectively, in Δ*pgmA* when cholesterol was used as a carbon source. To corroborate the above findings, we conducted a comparison of the relative expression levels of certain well-known adenylate cyclase genes of Mtb in Δ*pgmA* cultured in media with glycerol and cholesterol as the exclusive carbon sources. We noticed a significant upsurge in the expression levels of all adenylate cyclases in Δ*pgmA* grown in glycerol, compared to the wild-type strain (Fig. S6B). Surprisingly, in cholesterol-grown Δ*pgmA*, there was a significant decrease in the overall expression of the same genes relative to the wild-type strain (Fig. 6D). *prpD* gene of Mtb, which is known to be up-regulated in Mtb grown on cholesterol (42), was used as a control to monitor the overall expression of cholesterol-induced Mtb genes.

The efficient regulation of CCM of any pathogen is integral to its survival and growth of the pathogen inside the host. The same holds for Mtb as well. Studies on Mtb have already established an association between mycobacterial antibiotic tolerance and CCM (43). However, the implications of an altered CCM in Δ*pgmA* on the drug susceptibility profile of the pathogen have never been studied. To begin with, we estimated the MIC of H37Rv, Δ*pgmA*, and complemented strains under various growth conditions. Interestingly, we observed approximately ~13% and ~55% enhanced susceptibility of the Δ*pgmA* to rifampicin (RIF) and isoniazid (INH) relative to the wild-type and the complemented strains (Fig. 6E and F). Surprisingly, this phenotype was specific to cholesterol because we did not see any difference in the drug susceptibility profile within the strains

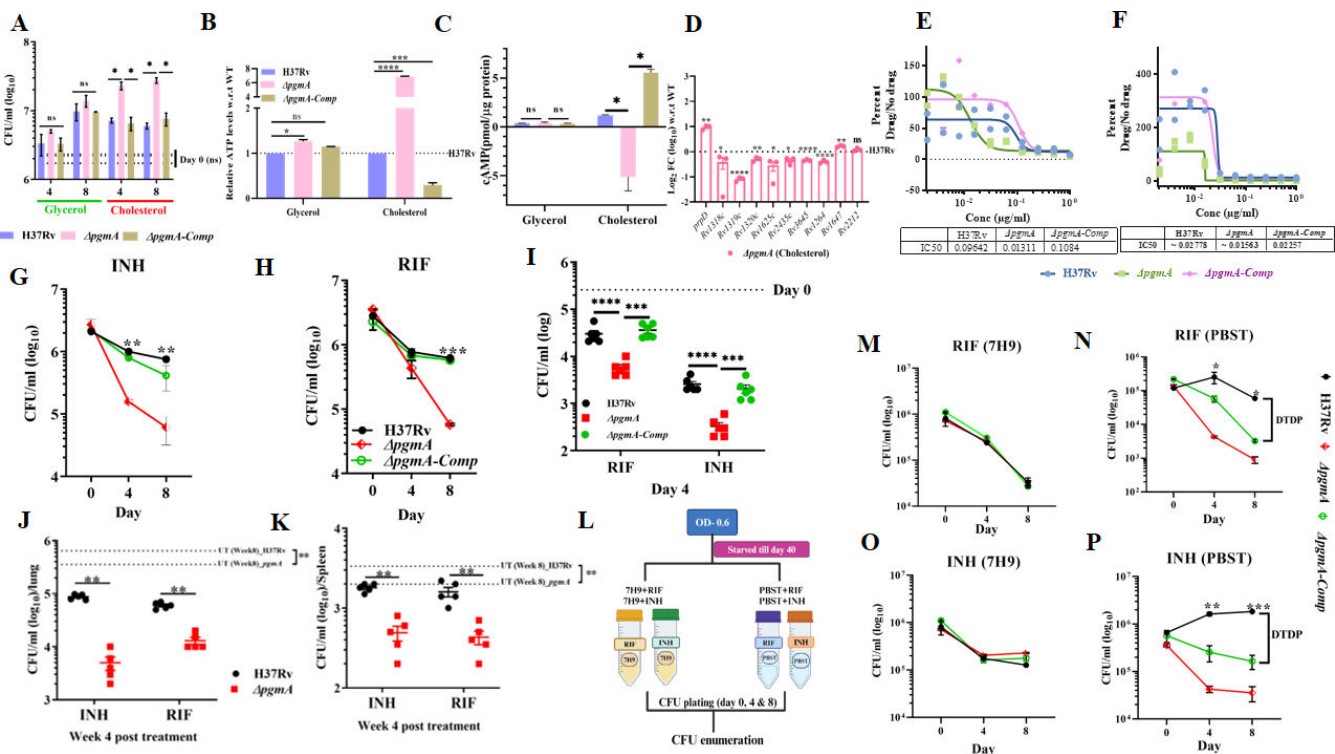

**FIG 6** *pgmA* is essential for cholesterol-specific modulation of growth and antibiotic persistence in Mtb. (A) Growth curve of H37Rv and Δ*pgmA* in a minimal medium supplemented with 0.1% glycerol and 0.01% cholesterol wherein Δ*pgmA* clearly overgrows in cholesterol-specific media. (B) cAMP and (C) ATP measurement of H37Rv and Δ*pgmA* in 0.01% glycerol and 0.01% cholesterol. (D) The expression levels of adenylate cyclases in the Δ*pgmA* were evaluated with respect to (w.r.t.) H37Rv under cholesterol-specific growth conditions. In (E) and (F), MIC determination was done using bioluminescence bacterial strains showing cholesterol media-specific susceptibility of *pgmA* to (E) isoniazid (INH) and (F) rifampicin (RIF), wherein below each drug response curve showed IC50 values in cholesterol media. (G) INH and (H) RIF-dependent time-kill assay of H37Rv, Δ*pgmA,* and Δ*pgmA-Comp* in cholesterol media. (I) THP-1 macrophages were infected with H37*Rv*, Δ*pgmA*, and Δ*pgmA-Comp* at an MOI of 1:10. After 4 h, 10× MIC of INH (0.3 µg/mL) and RIF (0.04 µg/mL) was added to the infected macrophages. At day 4 post-infection, macrophage cells were lysed, and bacillary survival was examined. CFU was obtained for drug-treated infected macrophages at day 4 post-infection. In (J) and (K), Balb/c mice (*n* = 5 per group at each time point) were aerosol challenged with H37*Rv* and Δ*pgmA*. After the establishment of infection for 4 weeks, antibiotics (INH or RIF) were given through oral gavaging for the next 4 weeks. A parallel group was left untreated (control). CFU enumeration was done from cell lysates of (J) lung and (K) spleen of Balb/c mice (*n* = 5) post INH and RIF treatment, wherein the untreated group (*n* = 5) was kept as a control. (L) Schematic representation showing experimental strategy for performing time-kill assay on starvation-induced dormant population of Mtb. (M) RIF and (O) INH dependent time-kill assay of the starvation-induced dormant population in 7H9 media. (N) RIF and (P) INH dependent time-kill assay of the starvation-induced dormant population in PBST media. Statistical significance, (A) to (D), (G), (H), and (M) to (P) was determined using unpaired, non-parametric, two-tailed *t*-tests, *$P \leq 0.05$, **$P \leq 0.005$, and ***$P \leq 0.0005$. Statistical significance of (I) was monitored using two-way analysis of variance wherein ***$P \leq 0.0001$, ****$P < 0.0001$. Statistical significance was determined using unpaired, non-parametric Mann-Whitney U-test in (J) and (K) **$P \leq 0.08$.Data represent mean ± SEM. DTDP, drug-tolerant dormant population.

under growth conditions where glycerol or an enriched media (7H9) was used as a carbon source (Fig. S6C through F). Furthermore, to study the role of the *pgmA* gene in inducing the drug tolerance phenotype in Mtb, we quantified the frequency of generation of persisters in different Mtb strains by performing a time-kill assay. Briefly, cells in the normal growth phase were subjected to INH and RIF at 10× MIC across various media specific to different carbon sources, followed by CFU plating on days 4 and 8. Similar to MIC data, we observed a cholesterol-specific decrease in the generation of persisters in Δ*pgmA* (Fig. 6H and I) with no observed difference in 7H9 and glycerol media (Fig. S6G through J). Furthermore, we validated the above *in vitro* data in both *ex vivo* macrophage and *in vivo* animal models. In the *ex vivo* study, THP-1 macrophages after infecting with H37Rv, Δ*pgmA,* and Δ*pgmA-Comp* were subsequently treated with 10× MIC of RIF and INH (Fig S6M). Interestingly, per the *in vitro* data, Δ*pgmA* exhibited increased susceptibility to both the first-line anti-TB drugs as compared to WT and complemented strains,

underscoring the indispensability of *pgmA* in drug-induced tolerance (Fig. 6J). These data further suggest that Mtb utilizes macrophage-derived cholesterol as one of the major nutrient sources inside the cell. To corroborate our *ex vivo* findings in an animal model, we administered drugs to groups of Balb/c mice infected with either the wild type or the Δ*pgmA* 4 weeks post-infection. The susceptibility of the strain to different drugs was estimated by CFU plating the lung and spleen lysates isolated from the drug-treated mice. Consistent with our findings from the macrophage study, the Δ*pgmA* strain was found to be susceptible to RIF and INH (Fig. 6K and L). Δ*pgmA* was found to be nearly 15% and ~10% more susceptible to INH and RIF, respectively, in the lungs and ~30% and ~40% in the spleen, respectively, when compared to the untreated group (Fig. S6L and M).

Furthermore, it is widely recognized that mycobacteria, when subjected to nutrient deprivation, enter a state of antibiotic tolerance or resistance (44, 45), a phenomenon primarily attributed to a number of factors such as membrane lipids like PDIMs, toxin-antitoxin systems, and transcription factors (46, 47). To investigate if the *pgmA* gene of Mtb contributes to this phenotype, we starved all three strains of Mtb for 40 days followed by treating each of these strains with RIF and INH at a concentration of 10× MIC under both nutrient-replete (7H9) and nutrient-depleted conditions (PBST) (Fig. 6M). Interestingly, the data from the time-kill assay revealed that starved cultures of the wild-type, Δ*pgmA*, and the complemented strains, when treated with both the drugs under nutrient-replete conditions, generated similar kill kinetics, suggesting the generation of an equal number of persisters (Fig. 6N and P). On the contrary, the starved wild-type strain was found to be completely tolerant to both the tested drugs. Intriguingly, this starvation-induced drug tolerance phenotype was completely dependent on the presence of the *pgmA* gene since the absence of this gene in Δ*pgmA* obliterated the generation of drug-tolerant bacteria, as evident in the time-kill assay (Fig. 6O and Q). Overall, the data suggest that *pgmA*-regulated CCM modulates drug susceptibility and is critical for inducing drug tolerance in the phenotype under nutrient-limiting conditions typically encountered by the pathogen inside the host.

## Deletion of *pgmA* impairs the growth of Mtb in mice

Finally, to elucidate the role of *pgmA* in mycobacterial virulence, we infected groups of C57BL/6 mice with wild-type, Δ*pgmA*, and the complemented strain and assessed virulence by measuring bacillary load via CFU plating at weeks 4 and 8 post-infection. A significant reduction in bacillary load was observed in the lungs and spleen of animals infected with Δ*pgmA* compared to those infected with H37Rv and the complemented strain (Fig. 7A and B). The reduction was more pronounced in the spleen. Additionally, upon gross examination, lungs and spleens isolated from Δ*pgmA*-infected mice exhibited reduced inflammation and fewer visible granulomas on the surface compared to organs from wild-type-infected mice (Fig. 7C). Histopathological analysis further demonstrated that the lungs of mice infected with Δ*pgmA* exhibited diminished disease pathology and a lower granuloma score relative to those infected with wild-type Mtb (Fig. 7D and E). Overall, these findings suggest that *pgmA* plays a crucial role in modulating Mtb virulence, and its absence appears to compromise the fitness and ability of Mtb to proliferate within the host.

## DISCUSSION

The phenotypic plasticity of Mtb is defined by its extraordinary ability to adapt to a variety of adverse environmental changes faced within the host. Although a stable central carbon metabolism ensures a continuous flow of energy and biomolecules essential for maintaining critical cellular processes, adaptability is defined by the ability of the pathogen to differentially regulate the initial flux of the available nutrient between these two pathways to maximize the efficiency of utilization of the available resources. Our data suggest that the conversion of G6P to G1P by *pgmA* is one such step that Mtb modulates to regulate the initial flux of carbon under nutrient-limiting and reactivation

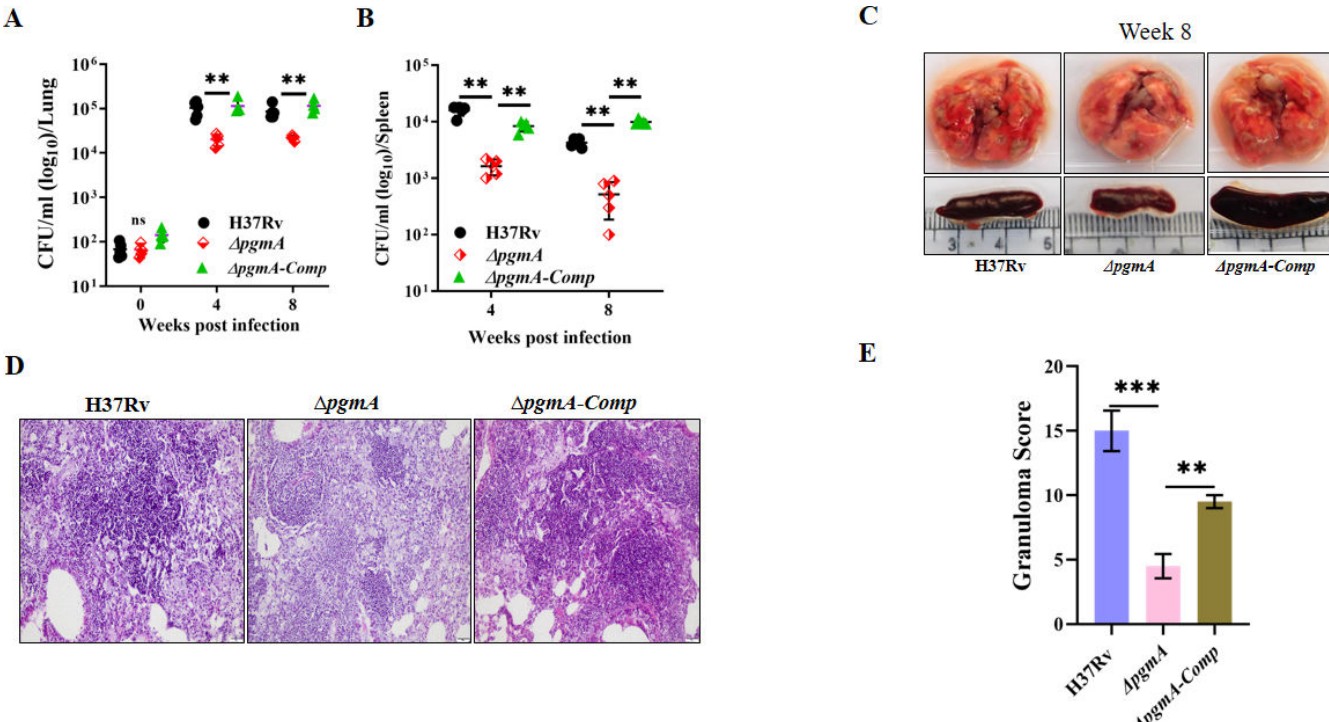

**FIG 7** Deletion of *pgmA* impairs the growth of Mtb in mice. Female C57BL/6 mice (*n* = 5) were subjected to aerosol challenge with H37Rv, Δ*pgmA,* and Δ*pgmA-Comp* strains to evaluate disease persistence in the (A) lungs and (B) spleen at specified time intervals. (C) Gross pathology of the infected lungs and spleen was assessed after 8 weeks post-infection. (D) Lung sections stained with hematoxylin and eosin were imaged at 20× magnification and analyzed for Mtb-induced histopathological changes. The alteration in lung morphology, including granuloma formation (E), is depicted. Statistical significance was determined using unpaired, non-parametric Mann-Whitney U-test in (A) and (B) **$P \leq 0.08$, and unpaired two-tailed *t*-test in (E), *$P \leq 0.05$, **$P \leq 0.005$, and ***$P \leq 0.0005$. Data represent mean ± SEM.

conditions. In this study, we found that the *pgmA*-mediated conversion of G6P to G1P or vice versa helps stabilize the CCM of Mtb by directing the carbon flux toward energy or biosynthesis depending on the growth condition. The reactivation kinetics demonstrate the importance of this flux in allowing Mtb to regulate the flow of carbon between different catabolic and biosynthetic pathways as and when needed. Interestingly, data from the growth and $^{13}C_3$ carbon-labeled dilution analysis suggest glycogen acts as a "carbon capacitor" and helps Mtb survive by releasing the stored carbon under nutrient-limiting conditions and during reactivation. Our lipid data reveal that the *pgmA*-dependent generation of G1P acts as a precursor for the metabolites essential for the synthesis and transportation of membrane-associated lipids critical for maintaining cell wall integrity and Mtb virulence (48, 49). Additionally, we report that *pgmA*-dependent growth modulation in cholesterol drives the generation of antibiotic persistence during tuberculosis infection, which can be further correlated with the cAMP levels as reported by recent reports (50, 51). Our study demonstrates that the *pgmA* gene of Mtb is indispensable for the long-term growth and survival of the pathogen inside the host.

Similar to our findings, using a transposon mutant library, it was recently reported that the *pgmA* gene is essential for Mtb to grow under both starvation and reactivation conditions (52). This phenotype is not restricted to mycobacteria and has been reported in several other bacterial pathogens like *Escherichia coli, Salmonella typhimurium*, and *Vibrio* spp., wherein mutants with impaired glycogen metabolism compromise fitness during carbon starvation (53, 54). The kinetics of starvation and reactivation is intriguing, suggesting that the *pgmA*-driven reaction is crucial for the adaptation of Mtb to an alternative nutrient source, particularly during the transition from nutrient-rich to nutrient-deprived conditions. This conclusion is supported by the initial lag observed at early time points, which disappears at later stages when the pathogen has adjusted

to utilizing alternative or available energy sources. Additionally, during this process, glycogen serves as a carbon capacitor, ensuring a stable energy supply during periods of starvation—a stability that is otherwise lost in the *pgmA* mutant. Furthermore, our data suggest that the *pgmA*-deficient mutant fails to divert the carbon flux into the synthesis of critical macromolecules essential for the generation of membrane-associated lipids, compromising the cellular morphology and membrane integrity and rendering the mutant strain susceptible to exposure to multiple stress conditions. The reported down-regulation of *pgmA* under hypoxia conditions is very intriguing (55). This, together with our findings, suggests that, despite showing a decrease in the transcript levels, *pgmA* is essential for the growth of bacteria under hypoxic conditions. Furthermore, our findings indicate that phosphoglucomutase-β (*Rv3400*), the protein of which has recently been studied (56), did not show redundancy upon removal of *pgmA*.

Defects in the formation of biofilm in Δ*pgmA* can be attributed to its inability to synthesize and transport Trehalose-dimycolates (TDM), also referred to as cording factor (57, 58). The importance of balanced and efficient regulation of CCM has been reported to be essential for the synthesis of mycolic acid critical for cell wall integrity and biofilm formation in Mtb (58–60). Mycolic acids are firmly anchored to arabinogalactans (AGs), forming a crucial component of the mycobacterial cell wall. The Δ*pgmA* mutant's inability to synthesize common UDP-linked sugar precursors, such as trehalose—an essential building block for AG biosynthesis—may contribute to the observed absence of surface-exposed mycolic acids. This hypothesis is further supported by our $^{13}C_3$ dilution analysis, which revealed a significant under-representation of trehalose and maltose-1-phosphate levels in Δ*pgmA* cells just before the onset of starvation, underscoring the metabolic disruption caused by *pgmA* deletion. The decrease in the levels of apolar membrane lipids like TAGs and a fraction of PDIMs from the cell membrane was very interesting. Lipid separation was performed during the late log and early stationary phases of cells, at which point it is known that Mtb switches to stored nutrients. TAG serves as a reliable long-term energy source with a lower molecular mass than glycogen (61). Its consumption in *pgmA* lacking strain indicates reliance on TAG in the absence of glycogen. Interestingly, the decrease in PDIM levels in Δ*pgmA* correlates with our RNAseq data, where we observed a Δ*pgmA*-specific down-regulation of *ppsD* (*Rv2934*) and *ppsB* (*Rv2932*) genes essential for the biosynthesis of PDIM in Mtb. Consistent with recent reports (62), we observed increased sensitivity of Δ*pgmA* to the peptidoglycan-targeting drug vancomycin, reinforcing the hypothesis that *pgmA* is essential for glycolipid-rich cell wall integrity of mycobacterium. The difference in the overall morphology, more specifically the difference in size, observed in Δ*pgmA*, we believe, is a strategy adopted by the Δ*pgmA* to remodel the cell wall to compensate for the lack of membrane glycolipids. The possibility of this being part of a strategy to conserve energy by the Δ*pgmA* cannot be ruled out. Similar observations have been reported in other bacterial species including *E. coli*, *Vibrio* spp., *Staphylococcus aureus*, and *Micrococcus luteus*, where cytoplasmic shrinkage occurs during dormancy to conserve energy (54, 63, 64). Alternatively, the failure of Δ*pgmA* to synthesize and store glycogen may limit ATP generation under nutrient-limiting conditions. Our RNAseq data provide clear evidence of major metabolic shifts in the absence of *pgmA*, blunting the ability of the pathogen to replicate and persist in the host during nutrient-depleting conditions. Furthermore, our data from the $^{13}C$ dilution study clearly demonstrate that disruption of glycogen synthesis in the Δ*pgmA* strain leads to an accelerated depletion of $^{13}C$-labeled metabolites (crucial for both growth and biosynthesis), indicating a reduced capacity to retain and recycle carbon. This underscores the critical role of stored glycogen in supporting Mtb's survival, particularly under nutrient-limiting conditions and during reactivation, where access to internal carbon reserves becomes essential for sustaining growth and persistence.

The *pgmA*-dependent modulation of Mtb growth in cholesterol was particularly stimulating. Although cAMP-dependent growth modulation in cholesterol is well established (65–68), our study is the first to implicate *pgmA* in modulating cAMP levels in

response to cholesterol. Given the emerging role of cAMP in Mtb drug susceptibility (67, 69–71). The cholesterol-specific carbon flux imbalance observed in Δ*pgmA* may explain the decreased cAMP and increased ATP levels, potentially underlying the overgrowth phenotype in cholesterol-rich conditions. Notably, the drug susceptibility profile of Δ*pgmA* under *ex vivo* and *in vivo* conditions was identical to that observed under *in vitro* cholesterol conditions, reinforcing the idea that host-derived cholesterol is a key nutrient source for Mtb survival. Whether the altered drug susceptibility profile of Δ*pgmA* stems from its increased growth rate or changes in cAMP levels remains unclear and needs further investigation.

It is well established that persisters exhibit tolerance to the majority of drugs (15, 44). Importantly, starvation-induced drug tolerance was completely abrogated in Δ*pgmA*, making the mutant more susceptible to drugs under non-permissive nutritional conditions. This suggests a dominant role of *pgmA* in regulating mycobacterium drug susceptibility phenotype under nutrient-limiting conditions. The specificity of this phenotype to nutrient deprivation underscores the significance of *pgmA*-regulated carbon flux in drug tolerance. Our ${}^{13}$C dilution study and colorimetric estimation of cAMP further confirm *pgmA*-specific metabolic shifts and reduced cAMP levels in nutrient-deprived conditions, establishing a link between cAMP regulation and drug susceptibility.

In conclusion, our study highlights the essential role of *pgmA* in Mtb survival, particularly during starvation and reactivation, by regulating CCM, cell wall integrity, antibiotic persistence, disease persistence, and cAMP-dependent growth modulation Fig. 8. These findings have significant implications for host-microbe systems biology, revealing how metabolic plasticity enables pathogen survival under host-imposed stress. However, as our study primarily focuses on bacterial metabolism, future investigations should explore how host immune responses influence PgmA activity and whether its inhibition affects host metabolic pathways.

While *pgmA* inhibitors show promise as adjunct therapies, their specificity and *in vivo* efficacy require further validation to minimize off-target effects. Understanding the interplay between *pgmA* function, host lipid metabolism, and immune evasion could provide deeper insights into tuberculosis pathogenesis and therapeutic development. Selectively targeting *pgmA* may enhance existing tuberculosis treatments, but successful integration into TB management will require the development of inhibitors that selectively target Mtb while preserving host metabolic balance. Thorough *in vivo* validation and integration with existing anti-TB drugs will be crucial to evaluate their therapeutic potential. If successfully implemented, *pgmA* inhibition could offer a novel strategy to counteract TB persistence and drug tolerance. However, further studies are needed to ensure that *pgmA*-targeted interventions do not adversely impact host metabolism.

## MATERIALS AND METHOD

### Bacterial strains, plasmids, and culture conditions

All the experiments were performed using the *Mycobacterium tuberculosis* H37Rv strain. The Δ*pgmA* (Δ*Rv3068c*) strain, as annotated in Mycobrowser, was generated by replacing the *Rv3068c* gene with a hygromycin resistance cassette using the pJM1 suicide vector to make *pgmA* knockout (Fig. S1). To study gene-specific effects, a complementation strain was generated by unmarking the deletion mutant. This whole cassette was excised using Cre recombinase. The unmarked mutant strain was then transformed with pMV261 harboring the *pgmA* gene under a constitutive mycobacterial *Phsp60* promoter. eGFP (enhanced Green Fluorescent Protein)-expressing strains were prepared by electroporating pMV261:eGFP plasmid. Transformants were selected on hygromycin-containing 7H11 agar plates, with green fluorescent colonies indicating successful plasmid uptake by the bacilli. The cultures were grown in Middlebrook 7H9 broth (Difco) supplemented with 0.05% Tween-80 (Sigma), 10% OADS (Oleic acid, Albumin, Dextrose, NaCl) enrichment, and 0.2% glycerol (Sigma) at 37°C, 100 rpm shaking. CFU plating was done on 7H11

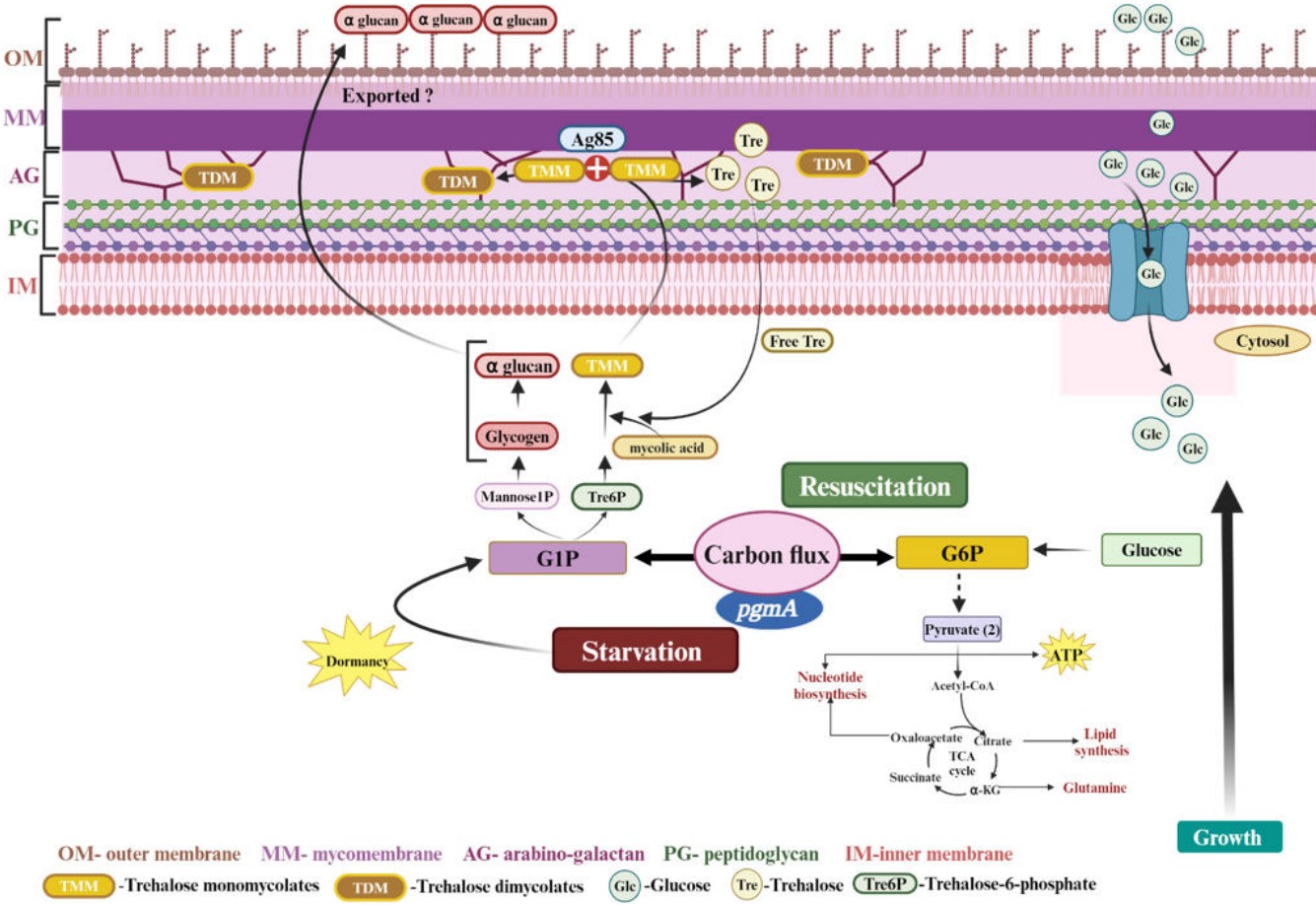

**FIG 8** Schematic representation of the pivotal role played by *pgmA* in regulating carbon homeostasis. *pgmA* orchestrates the conversion of G6P to G1P, thereby contributing significantly to stabilizing the CCM of Mtb. This regulation directs carbon flux toward either energy production or biosynthesis, contingent upon diverse growth conditions. Consequently, Mtb can thrive by tapping into stored carbon reservoirs during periods of nutrient scarcity and resuscitation phases. Moreover, the PgmA-mediated production of G1P acts as a critical precursor for metabolites essential in synthesizing membrane-associated lipids, pivotal for maintaining cell wall integrity. Hence, the PgmA-dependent carbon flux switch assumes indispensable importance for Mtb throughout disease progression and antibiotic tolerance. This flux facilitates adaptation to varying environmental conditions and ensures metabolic adaptability crucial for survival and virulence.

(agar) media supplemented with 10% OADS and 0.5% glycerol. Hygromycin and kanamycin were used at concentrations of 50 µg/mL and 25 µg/mL, respectively. Carbon source-specific experiments were done in minimal media (0.5 g/L asparagine, 1 g/L $KH_2PO_4$, 2.5 g/L $Na_2HPO_4$, 50 mg/L ferric ammonium citrate, 0.5 g/L $MgSO_4 \cdot 7H_2O$, 0.5 mg/L $CaCl_2$, 0.1 mg/L $ZnSO_4$) containing 0.1% (vol/vol) glycerol and 0.01% (wt/vol) cholesterol. For biofilm formation, Sauton's media was prepared using 0.5 g of $KH_2PO_4$, 0.5 g of $MgSO_4$, 4 g of L-asparagine, 2 g of citric acid, 0.05 g of ferric ammonium citrate, and 60 mL of glycerol in 900 mL of water followed by adjusting the pH to 7.0 with NaOH. The MIC determination and resuscitation dynamics were monitored using a biolumines-cence-based reporter strains haboring plasmid, pMV306hspLux+G13 (Addgene # 26161) (72). Whenever required, antibiotics were used as final with concentrations of 50 µg/mL (hygromycin) and 25 µg/mL (kanamycin and zeocin), respectively. Furthermore, the list of strains used in this study has been mentioned in Table S1. Also, the primers used in the study are enlisted in Table S2.

## Glycogen estimation assay

The glycogen estimation was performed using biochemical assay wherein, free glucose was measured across all the samples. Briefly, the bacterial cells were starved in PBST to

avoid the assay's limitation of background-free glucose interference present in 7H9. The estimation was done at day 7. To evaluate glycogen levels, cells were disrupted using zirconia beads in the assay buffer comprising 25 mM citrate at pH 4.2 and 2.5 g/L sodium fluoride, all maintained at 4°C. The cell lysates were clarified by centrifugation at 14,000 $g$ for 10 mins, the resulting supernatants were filtered and used for glycogen measurement by the EnzyChrom glycogen assay kit (Bioassay Systems), as per the instructions mentioned by the manufacturer.

### EtBr permeability assay

The cell wall permeability was assessed by EtBr dye and was adapted as originally mentioned earlier (73). Concisely, cells were harvested in a 7H9 complete medium without detergent until $A_{600}$ of 0.4–0.5. Following a phosphate-buffered saline (PBS) wash, cells were introduced into wells of a black, clear-bottom 96-well plate containing EtBr solution prepared in PBS at a final concentration of 5 µg/mL. Fluorescence readings were recorded every minute for 140 min at 530/590 nm, wherein PBS was kept as a control.

### Lipid extraction and thin-layer chromatography

Cultures were grown in 10 mL 7H9 media until the late log phase ($A_{600}$ ~1 to 1.5). The cells were then washed with PBS and resuspended in chloroform-methanol (2:1, vol/vol) mix overnight at 37°C and 100 rpm. This was followed by sequential extraction with chloroform-methanol (1:1, vol/vol) and chloroform-methanol (1:2, vol/vol). For analyzing lipid defects in biofilms, cells from the biofilm were taken and washed with PBS followed by sequential chloroform-methanol treatment. For lipid analysis in TLC, 10 µL of each lipid was spotted onto a pre-dried TLC plate (silica gel; Sigma) at a distance of 2 cm upward from the end of the plate. The solvent systems used for lipid analysis are listed in Fig. S3A.

### Transmission electron microscopy

Specimens for TEM analysis underwent processing according to established procedures outlined elsewhere (74). To summarize, cells were cultured in 7H9 broth (lacking Tween-80), subsequently starved in PBST for 7 days, and approximately $5 \times 10^7$ cells were harvested by centrifugation. The collected cells were then fixed using a combination of 2.5% paraformaldehyde and 2.5% glutaraldehyde. TEM was carried out utilizing a JEM 1400 Flash TEM operating at 120 kV (ATPC, Regional Center for Biotechnology, India), equipped with tungsten filament as an electron source and highly sensitive scientific Complementary Metal-Oxide Semiconductor (sCMOS) camera. The cell size was measured for $n = 50$ cells for each strain, and the analysis was done using RADIUS 2.0 EMSIS (build 14530) inbuilt software.

### Biofilm formation

Biofilms of each strain were cultivated in six-well plates by introducing 3 mL of Sauton's medium (excluding Tween-80 or tyloxapol) having 4% of fully saturated planktonic culture. These dishes were then placed in a stationary incubator at 37°C, under humid conditions with 5% $CO_2$, undisturbed for 5 weeks.

### *In vitro* stress assay

To determine the susceptibility to starvation and pH stresses, $A_{600}$ ~0.5–0.6 mycobacterial cultures were washed with PBS containing 0.01% Tyloxapol (PBST), and a single-cell suspension was inoculated at an $A_{600}$ of ~0.5 in PBST and ~0.2 in 7H9-ADC (pH 4.5). To assess viability after hypoxic stress, bacterial strains were inoculated with a headspace of 15% at an $A_{600}$ of ~0.1 in 7H9 medium containing 1.5 µg/mL methylene blue (colorimetric redox indicator of dissolved oxygen) in vacutainer vials at 37°C. For nitrosative and

oxidative stresses, 200 µM DETA-NO and 50 µM CHP were used. Bacterial samples were aliquoted at indicated time points and appropriately diluted, followed by CFU plating.

## Confocal laser microscopy

The eGFP-expressing strains were used for confocal microscopy experiments. THP-1 macrophages were seeded onto glass coverslips in a 24-well plate at a density of $5 \times 10^5$ cells per well and were activated with 10 ng/mL of human IFN-γ (Peprotech, USA) for 24 h before infection. Bacterial cultures were cultivated until they reached the early log phase ($A_{600}$ ~0.6–0.7), subsequently harvested, and rinsed with fresh RPMI-1640 media. Following this, a single-cell suspension was prepared using the soft-spin method (centrifugation at 120 $g$ for 10 min). The activated THP-1 cells were infected with an MOI of 1:10. After a 4 h incubation period, the cells were washed three times with pre-warmed PBS to eliminate extracellular bacteria, and the media (RPMI supplemented with 10% fetal bovine serum) was replenished. Subsequently, 24 h post-infection, the cells were stained with 50 nM LysoTracker Red DND-99 (Invitrogen Life Technologies, CA, USA) in complete RPMI media for 30 min at 37°C with 5% $CO_2$ and then fixed with 4% paraformaldehyde in PBS. Coverslips were mounted with ProLong Diamond antifade with 4′,6-diamidino-2-phenylindole (DAPI; Molecular Probes by Life Technologies, CA, USA). Finally, the slides were analyzed using an Olympus Confocal Laser Scanning Microscope. The presence of eGFP-expressing mycobacteria co-localizing with LysoTracker Red was determined by examining approximately 100 phagosomes. The CFU enumeration of all three strains was also determined by infecting activated and non-activated THP-1 macrophages with an MOI of 1:1 on day 6.

## Genome-wide expression analysis

Mtb cultures, conducted in triplicate, were harvested in 7H9 media until reaching OD: 0.6. Simultaneously, cultures from each strain (in triplicate) were starved (initial OD: 0.5) until day 4. Subsequently, cultures from both media conditions were resuspended in Trizol. The pelleted Trizol mix underwent bead beating (using 0.2 µM zirconia beads) five times for 40 s each, with 1 min intervals on ice to cool the sample. RNA was then isolated using the chloroform and isopropanol method. The total RNA was outsourced for subsequent DNase treatment and transcriptomic analysis. Raw reads were filtered using Trimmomatic to eliminate low-quality scores and adapters. The filtered reads were aligned to the Mtb reference genome (https://www.ncbi.nlm.nih.gov/datasets/taxonomy/1773/) using splice-aware aligners such as HISAT2 to quantify reads mapped to each transcript. The alignment percentage of reads ranged between 97.9% and 99.16% for all samples. The total number of uniquely mapped reads was counted using feature counts. The uniquely mapped reads were then subjected to differential gene expression analysis using DESeq2. Differential expression analysis of genes was performed for the starving condition versus the normal condition for both the *pgmA* mutant (*ΔRv3068c*) and H37Rv strains of Mtb, respectively, using a two-sample *t*-test. Genes with fold changes ±1 (log2) (25) and false discovery rates ≤0.05 were considered DEGs in the starving condition for both Δ*pgmA* and H37Rv, respectively.

## Liquid chromatography-mass spectrometry/metabolomics

Mtb cells were first harvested when the cultures attained OD: 0.6 in uniformly labeled (U)-13C3:12C glycerol at a concentration of 0.1% with 13C //12C, ¼: vol/vol. Furthermore, cells grown in 25% labeled glycerol were then transferred to starved media (PBST) with an initial OD of 0.5 until days 4 and 7. The starved cultures of mutant and parental strains were then washed twice with chilled 1× PBS. Bacterial cells were then lysed in extraction buffer (acetonitrile:methanol:water; 40:40:20) using 0.1 mm zirconia beads in a bead beater (five cycles/40 s pulse each). After each cycle, the cells were kept on ice. The lysed cells were pelleted by centrifugation at 5,000 rpm for 10 min at 4°C, and the supernatant was filtered twice using a 0.2 µm filter. The supernatant was dried using a

speed vacuum at room temperature and stored at −80°C until further analysis. Finally, for sample acquisition, the samples were resuspended in an equal ratio of methanol and water. The data acquisition was done on an Orbitrap Fusion mass spectrometer (Thermo Fisher Scientific) coupled with a heated electron ion source. The acquisition method has been followed as per the published protocols (75, 76) with minor modifications. The MS1 mass resolution was kept at 120,000, and for MS2, it was 60,000. Separation of the extracted metabolites was done on UHPLC Dionex Ultimate 3000. The data were acquired in both the reverse and hydrophilic interaction liquid chromatography (HILIC) phases in both positive and negative ionization modes. The reverse phase column was HSS T3, and the HILIC column was XBridge BEH Amide (Water Corporation). For the reverse phase, the mobile phase consists of solvent A (water + 0.1% formic acid) and solvent B (methanol + 0.1% formic acid). The elution gradient started with 1% B to 99% A over 10 min with a flow rate of 0.3 mL/min. For the HILIC, the mobile phase consists of solvent A (20 mM ammonium acetate in water) at pH 9.0 and solvent B (100% acetonitrile). The elution gradient started with 85% B to 15% A over 14 min with a flow rate of 0.2 mL/min. The injection volume was 5 µL. A quality control sample was run between the samples to monitor the signal variation and retension time (RT) shift. Data were analyzed using EL-MAVEN (v.0.12.1-beta) software, and ions were identified based on mass accuracy within 10 parts per million and 1 min retention time cut-off based on the in-house library of the metabolites of interest. The isotope natural abundance was corrected using AccuCor (77). To analyze isotopologs, the ratio of metabolite isotopic labeling was calculated by dividing the peak area ion intensities of total ion intensity of all labeled species of each metabolite with respect to the parental strain.

## ATP estimation

ATP levels were determined by pelleting approximately $10^8$ cells at 4,000 rpm. These cells were then washed twice with PBS and subsequently resuspended in 1 mL of PBS. Next, the cells were lysed using 0.1 mm zirconia beads in a mini-bead beater. Following this, the supernatant was collected after centrifuging the cell suspension at 14,000 rpm. The supernatant was further processed by heating it to 95°C and subsequently filtered through 0.2 µm filters. The resulting cell lysate was then subjected to ATP estimation, following the protocol by the manufacturer, utilizing the BacTiter-Glo Microbial Cell Viability Assay kit, Promega. Luminescence readouts were recorded and subsequently normalized per µg of protein.

## cAMP determination

To assess the cAMP levels within cells, approximately $10^8$ cell suspensions were subjected to centrifugation at 4,000 rpm. The resulting bacterial pellet was reconstituted in 1 mL of 0.1 M HCl, and the cells were disrupted using 0.1 mm zirconia beads in a mini Bead Beater from BioSpec Products, USA. Subsequently, beads and bacterial remnants were separated through centrifugation, and the resulting supernatants were employed for cAMP quantification using the manufacturer protocol provided in a direct immunoassay kit by Enzo (ADI-900-067A). The absorbance was taken at 405 nm, and the data were normalized per microgram of protein.

## RT-qPCR

Total RNA was isolated from the respective bacterial strains under specific conditions using Trizol (TaKaRa) following the protocol by the manufacturer. In brief, bacterial cells were washed with PBS and resuspended in Trizol. The cells were lysed by bead beating using 0.22 µm zirconia beads for five cycles, with 1-minute intermittent cooling steps between each cycle. Subsequently, RNA was extracted and precipitated using chloroform-isopropanol treatments. The resulting total RNA was treated with Turbo DNase I (Ambion) to remove DNA contamination. A total of 3,000 µg equivalent of DNase-treated RNA was used to generate cDNA using SuperScript IV Reverse Transcriptase (Invitrogen).

Finally, 1 µg equivalent of cDNA was utilized to set up the qPCR reaction with SYBR Premix Ex Taq by Takara Bio on QuantStudio 6 Flex. *sigA* was used as the internal control, and the ΔΔCt method was applied to determine the relative gene expressions concerning specified control conditions. *prpD* (*Rv1130*) was included as a positive control for cholesterol utilization.

## MIC and antibiotic tolerance experiments

The MIC was determined using bioluminescent bacterial strains with an OD of 0.5 (mid-log phase) that were washed with PBS. MIC values were determined in 7H9, glycerol, and cholesterol media. A 96-well plate was used for serially diluting the drugs, starting with twice the maximum concentration of the drug and media in the first well. Cells were added at a final density of OD 0.01 to the wells containing drugs and incubated at 37°C for 3 days. Luminescence was then measured, and dose-response curves were generated to calculate IC50 values as mentioned elsewhere with slight modifications (78, 79). Time-kill kinetics to assess *in vitro* antibiotic susceptibility were performed in different media - 7H9 and minimal medium supplemented with glycerol, and cholesterol as sole carbon source. Mid-log phase cultures were collected and transferred into their respective media until they reached an $A_{600}$ of approximately 0.2. A 10× MIC drug concentration was then added to these cultures, and CFU enumeration was performed on days 0, 4, and 8. For antibiotic susceptibility testing in THP-1 cells, the macrophages were infected at an MOI of 1:5. After a 4 h infection period, extracellular bacteria were removed through successive PBS washes. RPMI-1640 media containing a drug concentration of 10× MIC was then added to each well. Additionally, an untreated control group was maintained for each drug. CFU enumeration of intracellular bacterial load was monitored by cell lysis by PBST having 0.01% Triton-100 on day 4.

## Infection experiment in mice

For the pathogenesis study, C57BL/6 mice, aged 4 to 6 weeks, were housed in individually ventilated cages at the in-house biosafety level 3 facility. For the aerosol infection, groups of mice were exposed to bacterial suspension using the inhalation exposure machine, Glas col, LLC such that a deposition of 80–150 bacilli per mouse was achieved . At 24 h post-exposure, five mice per strain were sacrificed to assess bacterial deposition. To measure bacterial loads in the lungs and spleens at various specified time points, organs were aseptically harvested and homogenized in 2 mL of sterile normal saline. Subsequently, the homogenates were serially diluted and plated in triplicate on 7H11 agar with 10% OADS, supplemented with polymyxin B (50 mg/mL), amphotericin B (10 mg/mL), trimethoprim (10 mg/mL), vancomycin (5 mg/mL), carbenicillin (100 mg/mL), and cycloheximide (5 mg/mL) to prevent contamination and assess the bacillary load post incubation for 20-21 days. For the *in vivo* antibiotic efficacy study, BALB/c mice, aged 4–6 weeks, were used. For the establishment of infection, animals were kept under care for 4 weeks. After 4 weeks, *n* = 5 mice were sacrificed for bacillary load enumeration, and consequently, mice were randomly grouped and subjected to treatment with 10 mg/kg body wt. INH and RIF, respectively, through oral gavaging 5 days/week for 4 weeks. A parallel group was maintained as an untreated control. Bacillary survival was evaluated by sacrificing mice at 0, 4, and 8 weeks as mentioned earlier.

## Statistical analysis

Each *in vitro* and *ex vivo* CFU experiment was repeated thrice independently. The statistical significance was assessed using the Student's *t*-test in all the *in vitro* and *ex vivo* experiments and the Mann-Whitney U-test in animal experiments. Statistically significant results were defined as having values $P < 0.05$, $P < 0.005$, and $P < 0.0005$. All the analyses for RNAseq were performed on MATLAB and R software. Principle component analysis (PCA) was performed using the "prcomp" package of R. The "ttest2" function in MATLAB

was applied to perform the two-sample *t*-test. PCA plot, volcano plot, and heatmap were generated in R using ggplot2, gplots, and Complex Heatmap packages.

## ACKNOWLEDGMENTS

We acknowledge the funding from the Indian Council of Medical Research, India (project ID: EM/Dev/SG/212/7864/2023) to A.K.P. and from the Translational Health Science and Technology Institute (THSTI) core grant to A.K.P. and S.C. T.S. was supported by the Council of Scientific and Industrial Research (file no. 09/1049(0036)/2019-EMR-I) PhD fellowship. The sponsors had no involvement in the planning, data collection, analysis, interpretation, or in the preparation of the report.

Facility: We would like to acknowledge the support provided by the Biosafety level 3 (BSL3), MultiOMICS, Histopathology, and the Experimental Animal Facility (EAF) at THSTI.

Software: The graphical abstract depicting the findings was created using Biorender Software (https://www.biorender.com/). GraphPad Prism was employed for generating the plots.

Conceptualization: T.S. and A.K.P. Methodology: A.K.P., S.C., R.K.N., and Y.K. Experimentation: T.S., S.T., R.P., S.K.G., V.B., V.K.N., M.P., P.D., and B.N.P. Computational analysis: J.K. and S.C. Supervision: A.K.P. Writing-original draft: T.S. and A.K.P. Writing-review and editing: T.S., S.C., R.K.N., and A.K.P. The manuscript was read and approved by all the authors.

## AUTHOR AFFILIATIONS

[1]Mycobacterial Pathogenesis Laboratory, Centre for Tuberculosis Research, Translational Health Science and Technology Institute, Faridabad, Haryana, India

[2]Jawaharlal Nehru University, New Delhi, Delhi, India

[3]Complex Analysis Group, Computational and Mathematical Biology Centre, Translational Health Science and Technology Institute, Faridabad, Haryana, India

[4]Biomarker Discovery Laboratory, Non-communicable Disease Centre, Translational Health Science and Technology Institute, Faridabad, Haryana, India

[5]Experimental Animal Facility, Translational Health Science and Technology Institute, Faridabad, Haryana, India

[6]Translational Health Group, International Centre for Genetic Engineering and Biotechnology, New Delhi, Delhi, India

## AUTHOR ORCIDs

Taruna Sharma http://orcid.org/0009-0005-3434-9096
Vaibhav Kumar Nain http://orcid.org/0000-0002-7835-2044
Ranjan Kumar Nanda http://orcid.org/0000-0002-2163-4411
Amit Kumar Pandey http://orcid.org/0000-0002-6252-9060

## FUNDING

| Funder | Grant(s) | Author(s) |
|---|---|---|
| Indian Council of Medical Research | EM/Dev/SG/212/7864/2023 | Amit Kumar Pandey |
| Council of Scientific and Industrial Research, India | 09/1049(0036)/2019-EMR-I | Taruna Sharma |
| Translational Health Science and Technology Institute | Core Grant | Amit Kumar Pandey |
| | | Samrat Chatterjee |

## AUTHOR CONTRIBUTIONS

Taruna Sharma, Data curation, Formal analysis, Methodology, Validation, Visualization, Writing – original draft | Shaifali Tyagi, Data curation, Methodology | Rahul Pal, Data curation, Methodology | Jayendrajyoti Kundu, Formal analysis, Software, Visualization | Sonu Kumar Gupta, Formal analysis, Software | Vishawjeet Barik, Data curation, Formal

analysis, Methodology | Vaibhav Kumar Nain, Formal analysis, Methodology, Visualization | Manitosh Pandey, Formal analysis, Visualization | Prabhanjan Dwivedi, Formal analysis | Bhishma Narayan Panda, Formal analysis | Yashwant Kumar, Conceptualization, Formal analysis | Ranjan Kumar Nanda, Conceptualization, Formal analysis | Samrat Chatterjee, Conceptualization, Formal analysis | Amit Kumar Pandey, Conceptualization, Formal analysis, Funding acquisition, Investigation, Methodology, Project administration, Resources, Supervision, Validation, Visualization, Writing – review and editing

## DATA AVAILABILITY

The corresponding author had full access to all the study data and made the final decision to submit it for publication. All necessary data to assess the conclusions in the paper are provided either in the paper itself or in the supplementary materials. The RNA-Seq data sets have been deposited in the Indian Nucleotide Data Archive (Accession ID: PRJEB66126).

## ETHICS APPROVAL

The animal experimentation protocols were approved by the Institutional Animal Ethics Committee (IAEC), BRIC-THSTI with approval numbers - IAEC/THSTI/166 and IAEC/THSTI/211. The mice were cared for in strict accordance with established animal ethics and guidelines established by the Committee for Control and Supervision of Experiments on Animals (CCSEA), Government of India.

## ADDITIONAL FILES

The following material is available online.

### Supplemental Material

**Supplemental material (mSystems00420-25-s0001.pdf).** Figures S1 to S6 and Tables S1 to Table S4.

### Open Peer Review

**PEER REVIEW HISTORY (review-history.pdf).** An accounting of the reviewer comments and feedback.

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
