## [Reviewer comments · mSystems]

Phosphoglucosyltransferase A mediated metabolic adaptation is essential for antibiotic and disease persistence in *Mycobacterium tuberculosis*

Taruna Sharma, Shaifali Tyagi, Rahul Pal, Jayendrajyoti Kundu, Sonu Gupta, Vishawjeet Barik, Vaibhav Nain, Manitosh Pandey, Prabhanjan Dwivedi, Bhishma Panda, Yashwant Kumar, Ranjan Nanda, Samrat Chatterjee, and Amit Pandey

Corresponding Author(s): Amit Pandey, Translational Health Science and Technology Institute

Review Timeline:

Submission Date:	March 26, 2025
Editorial Decision:	May 8, 2025
Revision Received:	May 15, 2025
Accepted:	May 27, 2025

Editor: Sophie Darch

Reviewer(s): Disclosure of reviewer identity is with reference to reviewer comments included in decision letter(s). The following individuals involved in review of your submission have agreed to reveal their identity: Samuel Alvarez-Arguedas (Reviewer #3)

Transaction Report:

DOI: <https://doi.org/10.1128/msystems.00420-25>

Re: mSystems00420-25 (**Phosphoglucosyltransferase A mediated metabolic adaptation is essential for antibiotic and disease persistence in *Mycobacterium tuberculosis***)

Dear Dr. Amit Kumar Pandey:

After reading the reviewers comments i am satisfied that the revisions are sound and would like to move to accept your manuscript for publication. Reviewer 3 highlighted some minor modifications that after correction and resubmission, the paper will be routed as accepted.

Revision Guidelines

Sincerely,
Sophie Darch
Editor
mSystems

Reviewer #2 (Comments for the Author):

I thank the reviewers for answering all my questions and performing most of the experiments I have asked for. I believe the revised manuscript is now clearer and its quality is greatly improved. Well done.

Reviewer #3 (Comments for the Author):

The authors have addressed the changes requested by the first round of reviewers, and the manuscript is generally clear and well written. Below, I provide a few minor suggestions that may help to further improve clarity and consistency throughout the manuscript:

1- Figure-text correspondence (lines 150-151):

The description of bacterial length and diameter in the text appears in a different order than in the corresponding figure. Please adjust the text so that it matches the order presented in the figure.

2- Clarification of intracellular infection data (lines 173-174):

The manuscript notes that the Δ pgmA mutant shows reduced survival in both resting and activated macrophages (Figure 3B). As shown in Supplemental Figure 4B, the mutant strain does not appear to have impaired infectivity. Including this clarification in the main text would help underscore that the reduced intracellular survival is not due to altered infection efficiency.

3- Figure 4B presentation (lines 191-193):

I suggest removing the hits common to both wild-type and mutant strains in Figure 4B, as was done in Figure 4A. This would improve consistency between figures and facilitate interpretation.

4- Reference to Figures 7A and 7B (lines 344-345):

While discussing bacterial loads in lungs and spleens, Figures 7A and 7B are not referenced in the text. Please include appropriate citations to these figures. Also, consider standardizing the format of Figures 7A and 7B, as they represent the same type of data.

5- Units and formatting:

A careful revision of the units throughout the manuscript is recommended. For example:

Line 578: zirconia bead size is listed as 0.22 μ m - please confirm and maintain consistent formatting across the text.

Line 601: Filter pore size is noted as 0.2 μ m - check consistency with other instances.

Line 602: Please add the temperature units.

Line 616: The injection volume should be corrected to 5 μ l.

We sincerely thank the editor and all the reviewers for their time and thoughtful evaluation of our manuscript. We greatly appreciate the constructive feedback, which has significantly contributed to improving the scientific rigor and clarity of our work. Below, we provide a detailed, point-by-point response to the reviewer #3 comments and suggestions.

Reviewer #2 (Comments for the Author):

I thank the reviewers for answering all my questions and performing most of the experiments I have asked for. I believe the revised manuscript is now clearer and its quality is greatly improved. Well done.

We sincerely appreciate the time and effort Reviewer #2 put into the review process and are pleased to know that the revised version meets the expectations.

Reviewer #3 (Comments for the Author):

The authors have addressed the changes requested by the first round of reviewers, and the manuscript is generally clear and well written. Below, I provide a few minor suggestions that may help to further improve clarity and consistency throughout the manuscript:

1- Figure-text correspondence (lines 150-151):

The description of bacterial length and diameter in the text appears in a different order than in the corresponding figure. Please adjust the text so that it matches the order presented in the figure.

Corrected

2- Clarification of intracellular infection data (lines 173-174):

The manuscript notes that the Δ pgmA mutant shows reduced survival in both resting and activated macrophages (Figure 3B). As shown in Supplemental Figure 4B, the mutant strain does not appear to have impaired infectivity. Including this clarification in the main text would help underscore that the reduced intracellular survival is not due to altered infection efficiency.

Corrected

3- Figure 4B presentation (lines 191-193):

I suggest removing the hits common to both wild-type and mutant strains in Figure 4B, as was done in Figure 4A. This would improve consistency between figures and facilitate interpretation.

To enhance data transparency and facilitate direct comparison, the genes commonly differentially expressed in both the mutant and wild-type strains have been included in the volcano plot, as illustrated in Figure 4A. This addition allows for a clearer visualization of shared transcriptional responses and highlights the the number of DEGs that are consistently regulated across both backgrounds.

4- Reference to Figures 7A and 7B (lines 344-345):

While discussing bacterial loads in lungs and spleens, Figures 7A and 7B are not referenced

in the text. Please include appropriate citations to these figures. Also, consider standardizing the format of Figures 7A and 7B, as they represent the same type of data.

Corrected

5- Units and formatting:

A careful revision of the units throughout the manuscript is recommended. For example:
Line 578: zirconia bead size is listed as 0.22 μm - please confirm and maintain consistent formatting across the text.

Corrected

Line 601: Filter pore size is noted as 0.2 μm - check consistency with other instances.

Corrected

Line 602: Please add the temperature units.

Corrected

Line 616: The injection volume should be corrected to 5 μl .

Corrected

Re: mSystems00420-25R1 (**Phosphoglucosyltransferase A mediated metabolic adaptation is essential for antibiotic and disease persistence in *Mycobacterium tuberculosis***)

Dear Dr. Amit Kumar Pandey:

Your manuscript has been accepted, and I am forwarding it to the ASM production staff for publication. Your paper will first be checked to make sure all elements meet the technical requirements. ASM staff will contact you if anything needs to be revised before copyediting and production can begin. Otherwise, you will be notified when your proofs are ready to be viewed.

Sincerely,
Sophie Darch
Editor
mSystems